# CasExpress reveals widespread and diverse patterns of cell survival of caspase-3 activation during development in vivo

Xun Ding[1,2†], Gongping Sun[1†], Yewubdar G Argaw[1], Jessica O Wong[1], Sreesankar Easwaran[1], Denise J Montell[1,2*]

[1]Molecular, Cellular and Developmental Biology Department, University of California, Santa Barbara, Santa Barbara, United States; [2]Department of Biological Chemistry, Center for Cell Dynamics, Johns Hopkins School of Medicine, Baltimore, United States

**Abstract** Caspase-3 carries out the executioner phase of apoptosis, however under special circumstances, cells can survive its activity. To document systematically where and when cells survive caspase-3 activation in vivo, we designed a system, CasExpress, which drives fluorescent protein expression, transiently or permanently, in cells that survive caspase-3 activation in Drosophila. We discovered widespread survival of caspase-3 activity. Distinct spatial and temporal patterns emerged in different tissues. Some cells activated caspase-3 during their normal development in every cell and in every animal without evidence of apoptosis. In other tissues, such as the brain, expression was sporadic both temporally and spatially and overlapped with periods of apoptosis. In adults, reporter expression was evident in a large fraction of cells in most tissues of every animal; however the precise patterns varied. Inhibition of caspase activity in wing discs reduced wing size demonstrating functional significance. The implications of these patterns are discussed.

*For correspondence: denise.montell@lifesci.ucsb.edu

†These authors contributed equally to this work

**Competing interests:** The authors declare that no competing interests exist.

## Introduction

The cell death program known as apoptosis was originally described as a series of morphological changes that cells undergo as they die (*Tenev et al., 2005*). The reproducibility of the sequence suggested an underlying molecular program, and a conserved set of enzymes, the caspases, emerged as key regulators and executioners of apoptosis (*Martin and Green, 1995*; *Jacobson and Evan, 1994*; *Thornberry, 1998*). While caspase activation is frequently a terminal event resulting in swift cellular demise (*Chang et al., 2002*), cell survival following caspase activation has been described (e. g., [*Florentin and Arama, 2012*; *Kuranaga and Miura, 2007*; *Kumar, 2004*; *Meinander et al., 2012*]). In some cells and tissues, caspases promote localized or partial destruction of the cell without actually killing it (*Arama et al., 2003*; *Huh et al., 2004*; *Connolly et al., 2014)*.

A variety of primary cells and cell lines can survive caspase activation following a lethal dose of an apoptotic stimulus, as long as it is transient and thus sublethal in time (*Tang et al., 2012*). This reversal of late stage apoptosis has been named *anastasis* (Greek for 'rising to life'). Cell survival following caspase activation in response to a sublethal dose of irradiation has also been reported (*Florentin and Arama, 2012*; *Liu et al., 2015*; *Ichim et al., 2015*). Such survival following caspase activation has the potential for both beneficial and harmful effects. It may limit permanent damage to the heart following transient ischemia (*Kenis et al., 2010*); however it can also be oncogenic

**eLife digest** Every day, individual cells in our body actively decide whether to live or die. There are enzymes called executioner caspases that help cells to die in a carefully controlled process called apoptosis. Although the activation of executioner caspases generally leads to apoptosis, there are some circumstances in which cells are able to survive.

Fruit flies are often used in research as models of how animals grow and develop. Ding, Sun et al. set out to find more about the circumstances in which cells manage to survive caspase activation in fruit flies. The experiments used a new method that results in cells that survive caspase activity producing a fluorescent marker protein. This allowed Ding, Sun et al. to track when and where these events occurred in the flies.

Few cells in fruit fly embryos survive the activation of executioner caspase. However, in later stages of development, more and more cells survive this process. Cells in different parts of the body responded differently. For some types of cells, every cell seemed to survive caspase activity with no evidence of apoptosis. In other tissues like the central brain, in which a few cells normally choose to die, some cells occasionally managed to survive the activation of caspases. This rescue from the brink of death was more common than Ding, Sun et al. had anticipated.

The next step will be to uncover the molecular mechanisms that enable the cells to survive caspase activity. This knowledge may help us to develop treatments that can promote the survival of useful cells like heart muscle cells and brain cells, or trigger the death of cancer cells.

(*Tang et al., 2012*; *Liu et al., 2015*; *Ichim et al., 2015*), and could in principle allow tumor cells to escape chemotherapy.

Apoptosis is a critical feature of normal development in multicellular organisms (*Miura, 2012*; *Denton and Kumar, 2015*; *Vaux and Korsmeyer, 1999*). Studies in model organisms such as worms and flies have made important contributions to unraveling the underlying mechanisms (*Connolly et al., 2014*; *Denton and Kumar, 2015*; *Orme and Meier, 2009*; *Steller, 1995*). It is unknown whether cells ever recover from the brink of apoptotic cell death during development. The observations that cultured cells and adult cardiac myocytes recover from transient insults that cause caspase-3 activation raised the question as to how widespread cell survival following caspase activation might be in vivo, whether this ever occurs during normal development, and if so what function it might serve.

Identification of cells that survive transient caspase activation is challenging because they bear no known distinguishing characteristic. Therefore we developed a genetic system to mark and manipulate cells that survive caspase activation in Drosophila (*Figure 1*). Using these CasExpress transgenic flies, we discovered that the majority of cells in the adult derive from cells that survive caspase activation during normal development. We observed distinct categories of CasExpress activation. For example, in some organs, every cell activated the sensor over an extended period of development without evidence of apoptosis or morphological remodeling, suggesting a function for caspase-3 unrelated to cellular destruction. In other tissues, activation was sporadic in temporal and spatial pattern, suggesting a stochastic process. In these tissues, the precise patterns differed from animal to animal, and occurred in regions that normally exhibit apoptosis. These observations suggest that some cells recover from the brink of apoptotic cell death and undergo developmental anastasis. We propose that these different patterns represent distinct functions of executioner caspases during normal development.

## Results

### Design of CasExpress, an in vivo sensor for cells that survive caspase activation

In order to detect and follow the fates of cells that survive caspase activation, we designed a caspase-inducible Gal4 transcription factor (*Figure 1A*). To keep Gal4 inactive in the absence of caspase activity, we tethered it to the plasma membrane by fusing it to mCD8 (mouse cluster of

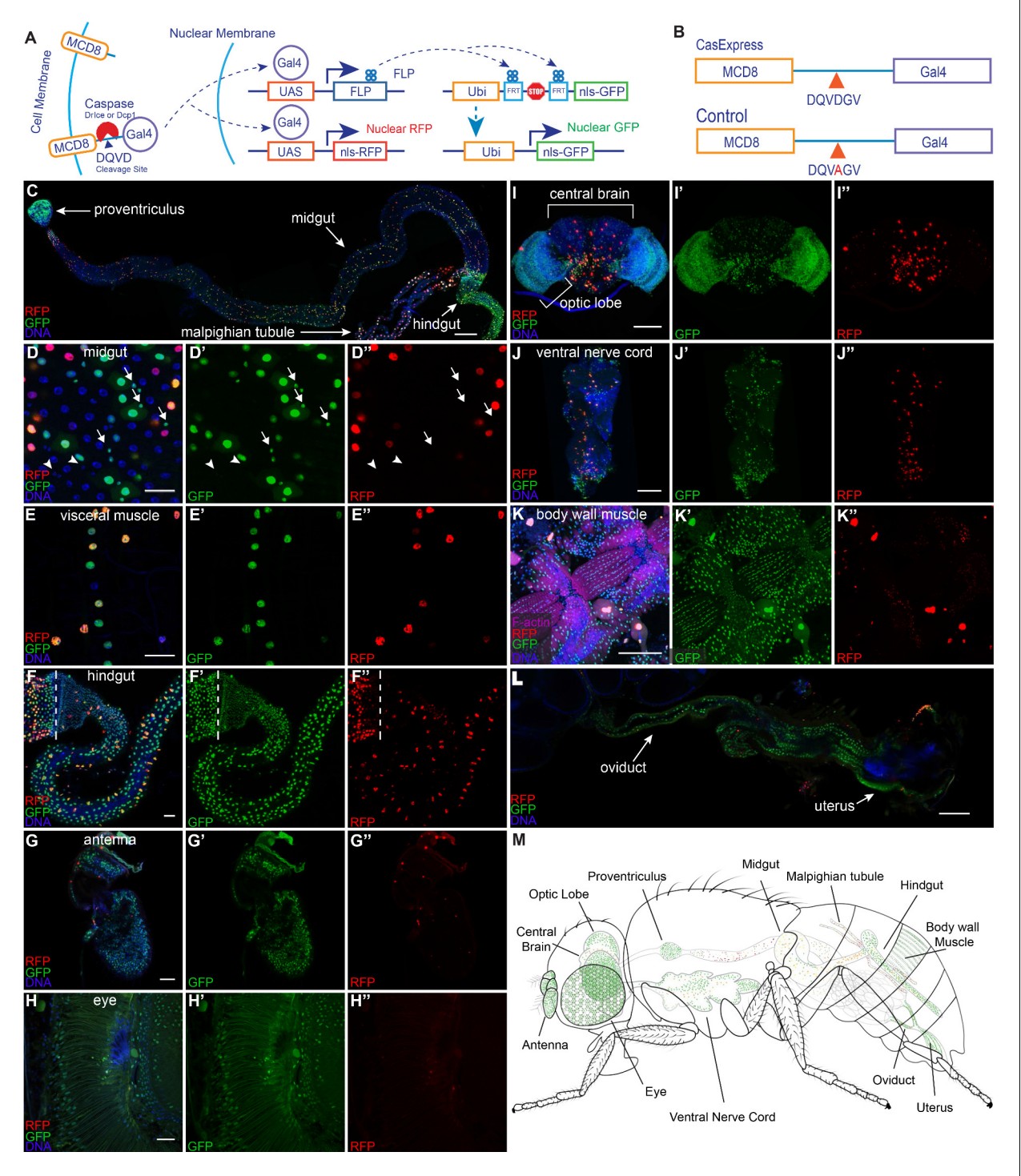

**Figure 1.** Widespread CasExpress activation in adult tissues. (**A**) A schematic of CasExpress and G-trace. (**B**) A schematic showing the sequence of the DQVD caspase cleavage site in CasExpress and the point mutation in the DQVA control. (**C–L**) Confocal micrographs showing overlays of DAPI, RFP and GFP from CasExpress/G-Trace flies. (**D'–L'**) GFP channel only. (**D''–L''**) RFP channel only. Arrows in **D–D''** indicate examples of GFP+ progenitor cells, and arrowheads point to examples of GFP- progenitor cells. Dotted lines in **F–F''** mark the boundary between midgut and hindgut. Scale bars in C and I-L are 100 μm; scale bars in **D–H** are 25 μm. (**M**) A schematic summarizing the general pattern of GFP and RFP expression in adult. Although GFP expression was present in all body wall muscle, only part is shown in green for simplicity and presentation clarity.

differentiation 8). To render the protein caspase-inducible, we inserted the caspase-3-binding and cleavage domain from the Drosophila Inhibitor of Apoptosis Protein 1 (DIAP1) (*Ditzel et al., 2003*) in between CD8 and Gal4. As a negative control we created a second transgene with a DQVD to DQVA amino acid substitution in the caspase cleavage site (*Figure 1B*) in order to render it caspase insensitive, hereafter the 'DQVA control.' To allow for detection of caspase activation in as many cell types as possible, the fusion protein was expressed under the control of the ubiquitin (ubi) enhancer/promoter. We characterized the expression and activity of transgenic flies bearing a site-directed insertion into the attP40 landing site, selected for its ability to allow relatively uniform, moderate levels of expression in a variety of tissues (*Markstein et al., 2008*). We also generated an insertion into a random site for comparison. We named this system CasExpress for its ability to drive expression of downstream genes and proteins under the control of caspase-3 activity.

## Widespread activation of CasExpress in the adult

To detect caspase activity, we crossed the sensor and control to G-Trace (*Evans et al., 2009*) a fly line that expresses two fluorescent protein targets, under the control of Gal4-responsive UAS (upstream activating sequences). G-Trace flies contain three transgenes, all on the second chromosome: UAS-RFP, UAS-FLP, which encodes a yeast recombinase enzyme, and a ubi-FRT-STOP-FRT-GFP cassette where FRT stands for FLP Recombination Target sequence. Crossing the mCD8-DQVD-Gal4 sensor to G-Trace should lead to permanent GFP expression in any cell that survives transient caspase activation and in all of its progeny, in contrast to other caspase activity reporters (*Bardet et al., 2008*). We expected the caspase-activated Gal4 protein to be short-lived because we had observed rapid degradation of other caspase reporters (*Tang et al., 2012*), so we anticipated RFP would be transient and limited to the cells that activated caspase-3 but not their progeny.

We first examined adult tissues where, to our surprise, we found widespread GFP expression (*Figure 1C–L*). In the intestine for example, GFP was evident in the most anterior structure, the proventriculus (*Figure 1C*), although little RFP was evident there, suggesting that caspase had been active earlier during development. In the midgut both RFP and GFP appeared in a partially overlapping pattern (*Figure 1C,D–D"*). Large nuclei corresponding to differentiated epithelial cells expressed both RFP and GFP suggesting ongoing caspase activation, whereas a subset of small progenitor cells expressed GFP but not RFP (*Figure 1D–D"* arrows). Visceral muscle and hindgut showed a mixture of GFP+/RFP- cells as well as some GFP+/RFP+ cells (*Figure 1E–F"*). The adult eye and antenna exhibited widespread nuclear GFP but only infrequent RFP (*Figure 1G–H"*), suggesting that caspase had been activated earlier in development either in a large fraction of cells, or in precursors that gave rise to a large fraction of adult cells; however little activation of caspase appeared to be ongoing in the adult.

In the adult central brain and nerve cord, a minority of cells expressed GFP and/or RFP (*Figure 1I–J"*). In the optic lobe, many but not all cells expressed GFP and/or RFP (*Figure 1I–I"*), whereas in body wall muscle, nearly every cell expressed GFP (*Figure 1K–K"*). In the female reproductive system, every cell of the oviduct was GFP+/RFP- in every animal, whereas the majority of germline and somatic cells in egg chambers lacked FP expression (*Figure 1L*). *Figure 1M* summarizes these findings schematically. Recently a similar strategy detected similarly widespread adult expression (*Tang et al., 2015*).

## Distinct spatial and temporal patterns of CasExpress during development

The adult expression suggested that CasExpress was activated during development. To document when caspase activation first appeared, we examined embryonic and larval stages. In Drosophila embryos, the only tissue that activated CasExpress robustly was the salivary gland beginning at stage 12 (*Figure 2A–A"*). Salivary gland expression was not detected in the DQVA control, demonstrating that this was not due to leaky or background expression from the G-Trace transgenes or random breakdown of the fusion protein that might separate Gal4 from the transmembrane domain. We also confirmed that the DQVD sensor and DQVA control showed similar patterns and levels of fusion protein expression at the cell surface detected with anti-mCD8 antibody staining throughout the embryo and in most tissues and stages of development (*Figure 2—figure supplement 1*). In the embryo RFP was also detected in some randomly distributed cells, likely corresponding to a subset

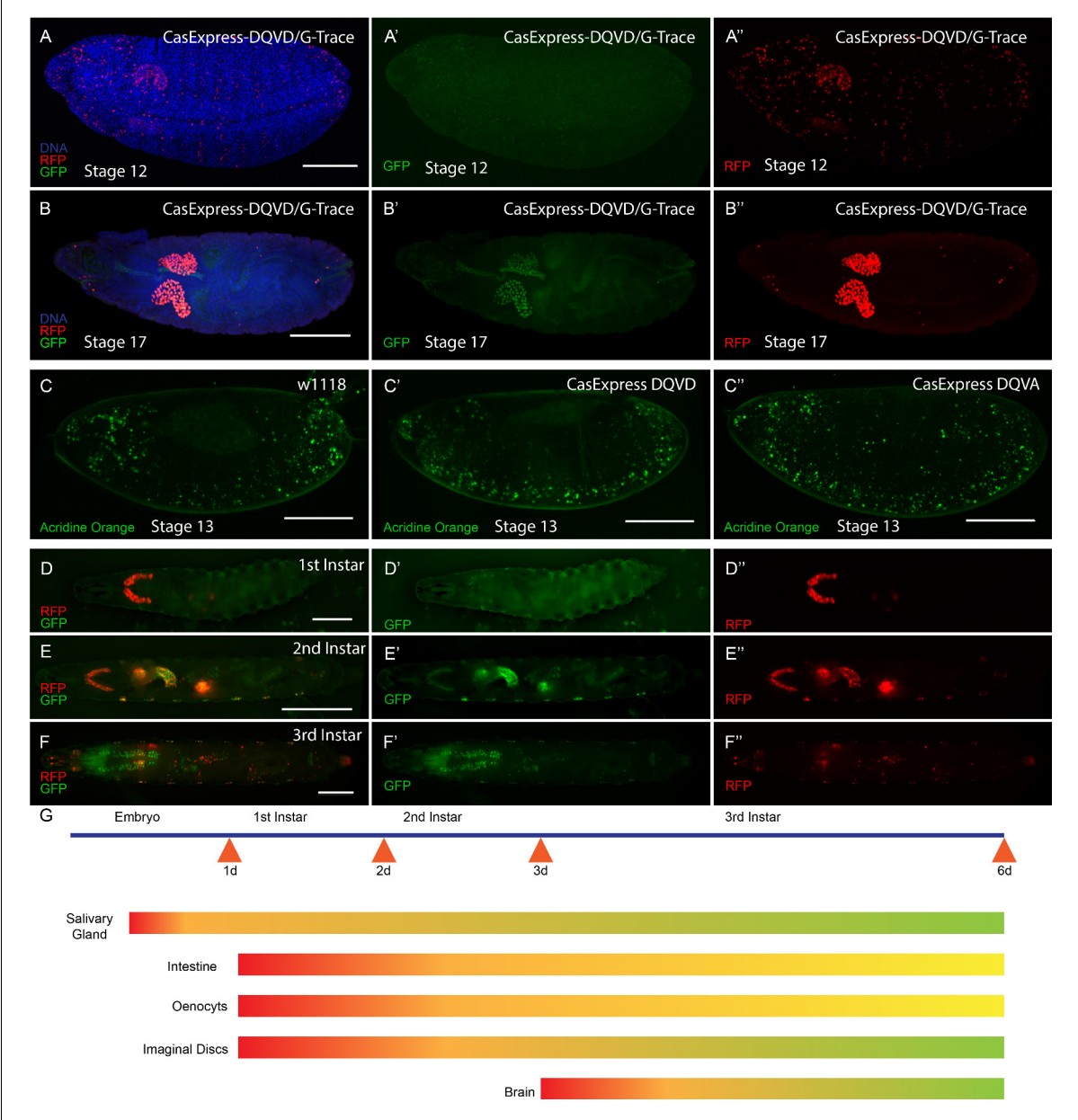

**Figure 2.** CasExpress activation in embryos and larvae. (A–B") RFP and GFP expression in Drosophila embryos (A–A") stage 12, (B–B") stage 17. (C–C") Acridine orange detection of apoptotic cells in stage 13 embryos of the indicated genotypes. (D–D") 1st instar larva, (E–E") 2nd instar larva and (F–F") 3rd instar larva. (G) A schematic summarizing of GFP and RFP expression in above stages. Red represents RFP expression. Green represents GFP. Yellow/Orange indicates either a mixture of GFP positive and RFP positive cell populations or the presence of cells expressing both. Scale bars represent: 100 μm (A–C); 200 μm (D); 400 μm (E); and 600 μm (F).

The following figure supplement is available for figure 2:

**Figure supplement 1.** Expression of CasExpress DQVD and the DQVA control.

of cells that normally undergo apoptosis (*Figure 2A and A"*); little if any GFP was detected in those cells, presumably because dying cells were not active enough to transcribe and translate FLP, undergo DNA recombination, and then transcribe and translate GFP to detectable levels. While RFP was detected in the salivary gland beginning in stage 12, GFP expression became evident later (*Figure 2B–B"*), confirming that these FPs exhibit different timing of activation.

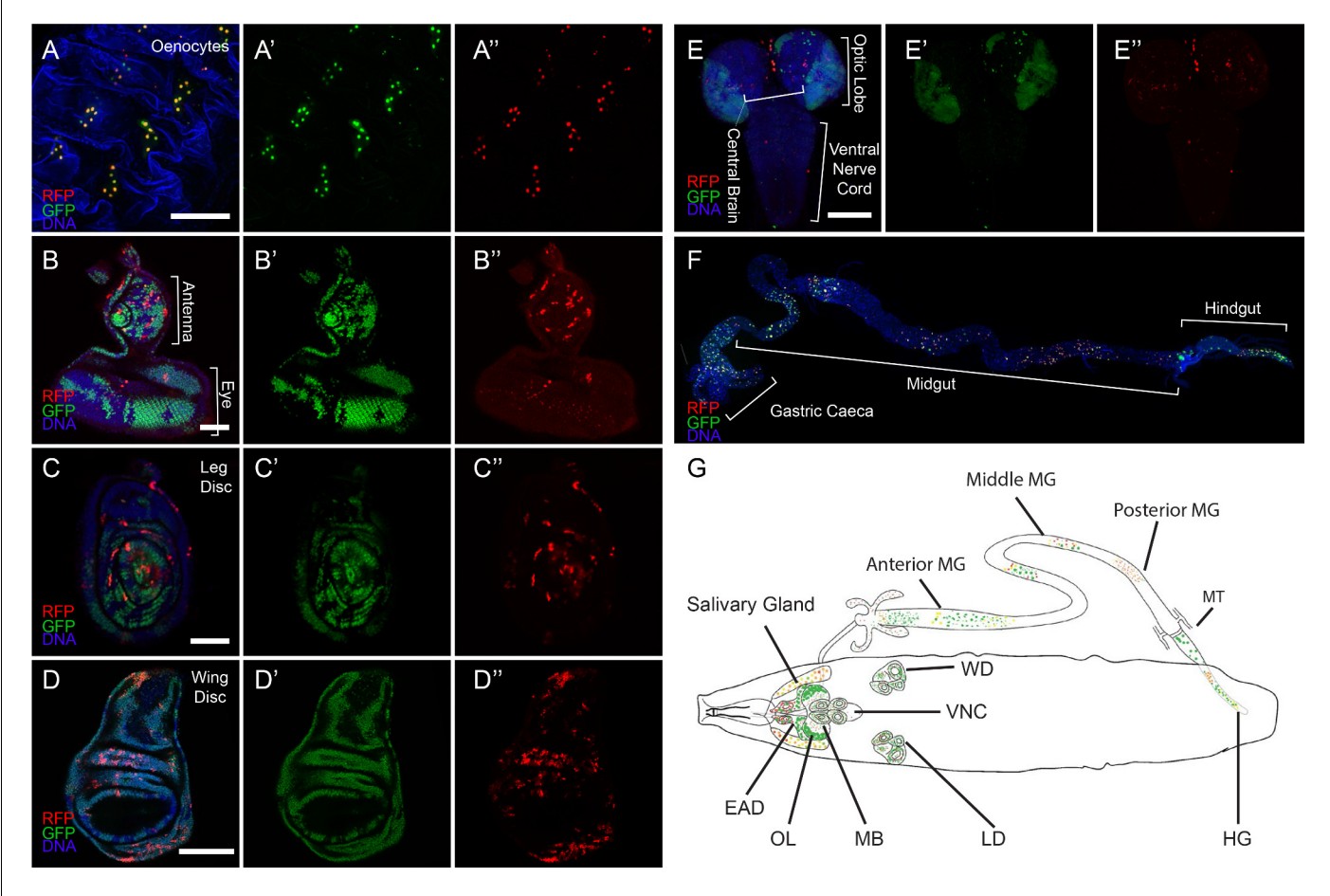

**Figure 3.** CasExpress activation in larval tissues. (**A–F**) Confocal micrographs showing overlays of DAPI, RFP and GFP expression in the indicated tissues of wandering 3rd instar larvae. (**A'–E'**) GFP only. (**A"–E"**) RFP only. The brackets in **D** mark the eye and antenna parts of the disc, in **E** mark the position of optic lobe, central brain and ventral nerve cord, and in **F** mark the different regions of the gut. Scale bars in **A** and **F** are 200 μm, in **B** and **C** are 50 μm, in **D** and **E** are 100 μm. (**G**) A schematic summarizing of GFP and RFP expression in larvae. There is little GFP/RFP expression in trachea or muscles, which are not included in diagram.

The following figure supplements are available for figure 3:

**Figure supplement 1.** Comparison of GFP expression in the CasExpress DQVD sensor, the DQVA control or G-trace alone.

**Figure supplement 2.** Diverse CasExpress activation patterns in larval CNS.

**Figure supplement 3.** Diverse CasExpress patterns in leg and wing discs.

**Figure supplement 4.** CasExpress patterns in larval intestines.

Although the DQVD and DQVA proteins contained the caspase binding sequence from DIAP1 (*Tenev et al., 2005*), which in principle could function as a dominant-negative inhibitor of caspase activity if expressed at high enough levels, the flies expressing the sensor were viable and fertile and showed no discernible morphological defects. The modest expression level and membrane localization presumably prevented any dominant negative effect. Moreover, there was no decrease in the number, or change in distribution, of apoptotic cells in DQVD and DQVA embryos compared to w[1118] embryos (*Figure 2C–C"*).

Both RFP and GFP continued to be expressed throughout embryonic and larval development (*Figure 2D–F"*). The temporal appearance of RFP and GFP in embryonic and larval life are indicated schematically in *Figure 2G*.

During larval development CasExpress activation appeared over time in many cell types and tissues including all imaginal discs, oenocytes, and in subsets of neurons (*Figure 2E* and *Figure 3*). Tissues from flies carrying G-Trace in the absence of the caspase sensor or in combination with the DQVA caspase-insensitive control exhibited infrequent FP expression in small clones in a minority of animals (*Figure 3—figure supplement 1*). The frequency and patterns were very similar regardless of the presence or absence of the DQVA control transgene (*Figure 3—figure supplement 1*), suggesting that this minor background was due to leaky, Gal-4-independent FLP expression from the UAS-FLP transgene. In contrast, expression in the presence of the DQVD caspase-sensitive construct was present in every animal (*Figure 3—figure supplement 1*), and in large fractions of cells (*Figure 3*).

Different tissues exhibited distinct temporal and spatial patterns. For example oenocytes exhibited RFP and GFP expression in virtually every cell and in every animal (*Figure 3A–A"*). In contrast, in imaginal discs fewer cells expressed RFP as compared to GFP (*Figure 3B–D"*). Although every disc from every animal exhibited expression, the precise patterns varied (*Figure 3—figure supplements 2–4*). In the developing central nervous system (CNS) the patterns were not bilaterally symmetric (*Figure 3—figure supplement 2*). In the imaginal discs the patterns, particularly of RFP, varied from animal to animal and did not appear to correspond to known developmental patterns of known signaling pathways or cell types (*Figure 3—figure supplement 3*).

Tissues that showed little or no activation of the sensor during normal development up through the third instar included somatic muscles, trachea, and the ventral nerve cord (*Figure 3E–E"*). Although most of the nervous system showed little sensor activation, a consistently large fraction (50–80%) of cells in the developing optic lobes were GFP-positive (*Figure 3E–E"*). In the larval intestine, partially overlapping GFP and RFP expression patterns were observed (*Figure 3F*), and while the overall regional patterns were conserved from one animal to the other, the details varied (*Figure 3—figure supplement 4*). The third instar larval patterns are summarized schematically in *Figure 3G*.

## Caspase dependence of the sensor

The unexpectedly widespread activation of CasExpress raised the question as to its caspase-dependence. The sensor inserted into the attP40 site and the random insertion demonstrated similar patterns. The absence of expression in the DQVA control demonstrated that a proteolytic cleavage at the aspartic acid was likely necessary. To address the possibility that a protease other than caspase activated CasExpress, we crossed the sensor and G-Trace into a homozygous *dronc* mutant background. Dronc encodes the upstream apoptotic caspase in Drosophila (equivalent to caspase-9 in mammals, (*Meier et al., 2000*; *Hawkins et al., 2000*) and its activity is necessary for activation of both fly executioner caspase molecules Drice and Dcp-1 (*Florentin and Arama, 2012*; *Song et al., 1997*; *Fraser et al., 1997*; *Fraser and Evan, 1997*; *DeVorkin et al., 2014*; *Muro et al., 2006*). Although Dronc mutants are homozygous lethal, they survive to the third instar larval stage allowing us to assess CasExpress at that stage. As expected, the homozygous *dronc* mutant background eliminated caspase activity detected with an antibody against cleaved and activated Dcp-1 (c-Dcp-1) (*Figure 4A–F'*). The *dronc* mutant also eliminated virtually all RFP and GFP expression in imaginal discs from CasExpress (*Figure 4A–F*).

Homozygous *dronc* mutant embryos retained RFP and GFP expression in the salivary gland, possibly due to the perdurance of maternal caspase expression. To confirm the presence of cleaved caspase in embryonic salivary gland cells, which has not been previously reported, we stained CasExpress embryos with an antibody against cleaved caspase-3 (*Figure 4—figure supplement 1*). Despite the absence of other apoptotic markers in these cells, salivary glands did label with this antibody, suggesting a non-apoptotic function for caspase in this tissue.

The Baculovirus p35 protein inhibits both executioner caspases DrIce and Dcp-1, but not Dronc (*Meier et al., 2000*). Therefore we crossed GMR-p35, which is a transgene that expresses p35 in the eye imaginal disc posterior to the morphogenetic furrow (*Hay et al., 1994*), into the CasExpress/G-Trace flies. GMR-p35 significantly reduced the number of RFP+ cells in the posterior eye disc compared to the control (*Figure 4 G-I*), whereas no change in RFP was observed in the antennal disc,

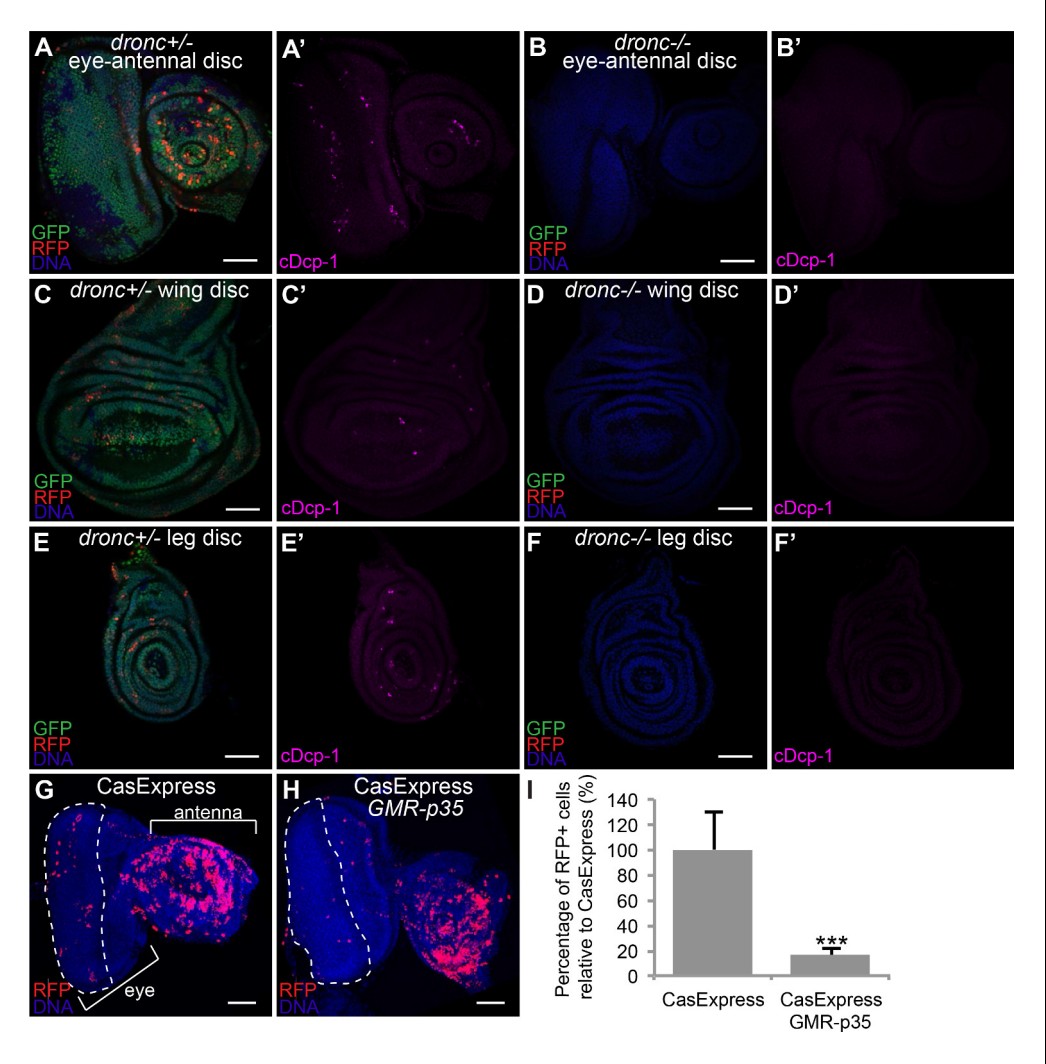

**Figure 4.** Caspase-dependence of CasExpress. (A–F) Confocal micrographs showing overlays of DAPI, RFP and GFP expression in third-instar larval eye-antennal disc (A–B), wing disc (C–D), and leg disc (E–F). CasExpress and G-trace were crossed into heterozygous (A, C, E) or *dronc* homozygous (B, D, F) *dronc* mutants. (A'–E') Cleaved Dcp-1 staining of corresponding discs. Scale bars are 50 μm. (G–H) RFP expression in eye-antennal discs of late third-instar larvae with CasExpress and G-trace with (H) or without (G) GMR-p35. The dashed line encircles the region where p35 is expressed. (I) Quantification of of RFP: DAPI area. Error bars show standard error of the mean, and *** indicates p<0.001.

The following figure supplements are available for figure 4:

**Figure supplement 1.** Anti-cleaved caspase-3 (red) and DAPI (blue) staining of stage 14 embryo (A) and high magnification of a salivary gland (B).

**Figure supplement 2.** GFP and RFP expression of CasExpress in a wild type eye-antennal disc (A–A"), or one carrying the GMR-p35 transgene (B–B").

**Figure supplement 3.** Loss of dredd does not change CasExpress patterns.

which served as an additional internal control. The few remaining RFP+ cells in the posterior eye disc likely were cells that had activated Gal4 prior to the onset of expression of the GMR promoter. GFP expression was still evident, indicating that caspase activation preceded expression of p35 from the GMR enhancer/promoter in those cells (*Figure 4—figure supplement 2*).

One known non-apoptotic role for caspase activity is in the innate immune response. Specifically the upstream caspase Dredd activates NFkB signaling and expression of anti-microbial peptides (*Meinander et al., 2012*; *Leulier et al., 2000*). The gut is known to have a highly active innate immune response. Therefore to determine whether the CasExpress activity we detected in the gut was due to the immune response, we crossed CasExpress into *dredd* mutant animals. However we detected no difference in the GFP or RFP expression level or pattern between *dredd* mutants and heterozygous wild type siblings, in any tissue examined (*Figure 4—figure supplement 3*).

## Developmental timing of caspase activation

To address when the CasExpress was activated in various tissues, we silenced CasExpress during most of development, by crossing in the temperature-sensitive (ts) version of Gal80 (Gal80$^{ts}$), which represses expression from UAS transgenes even in the presence of Gal4. When flies carrying CasExpress, Gal80$^{ts}$, and G-TRACE were grown at 18°C, GFP was completely repressed, and even the infrequent, random clones due to leaky expression of UAS-FLP was suppressed (*Figure 5—figure supplement 1*).

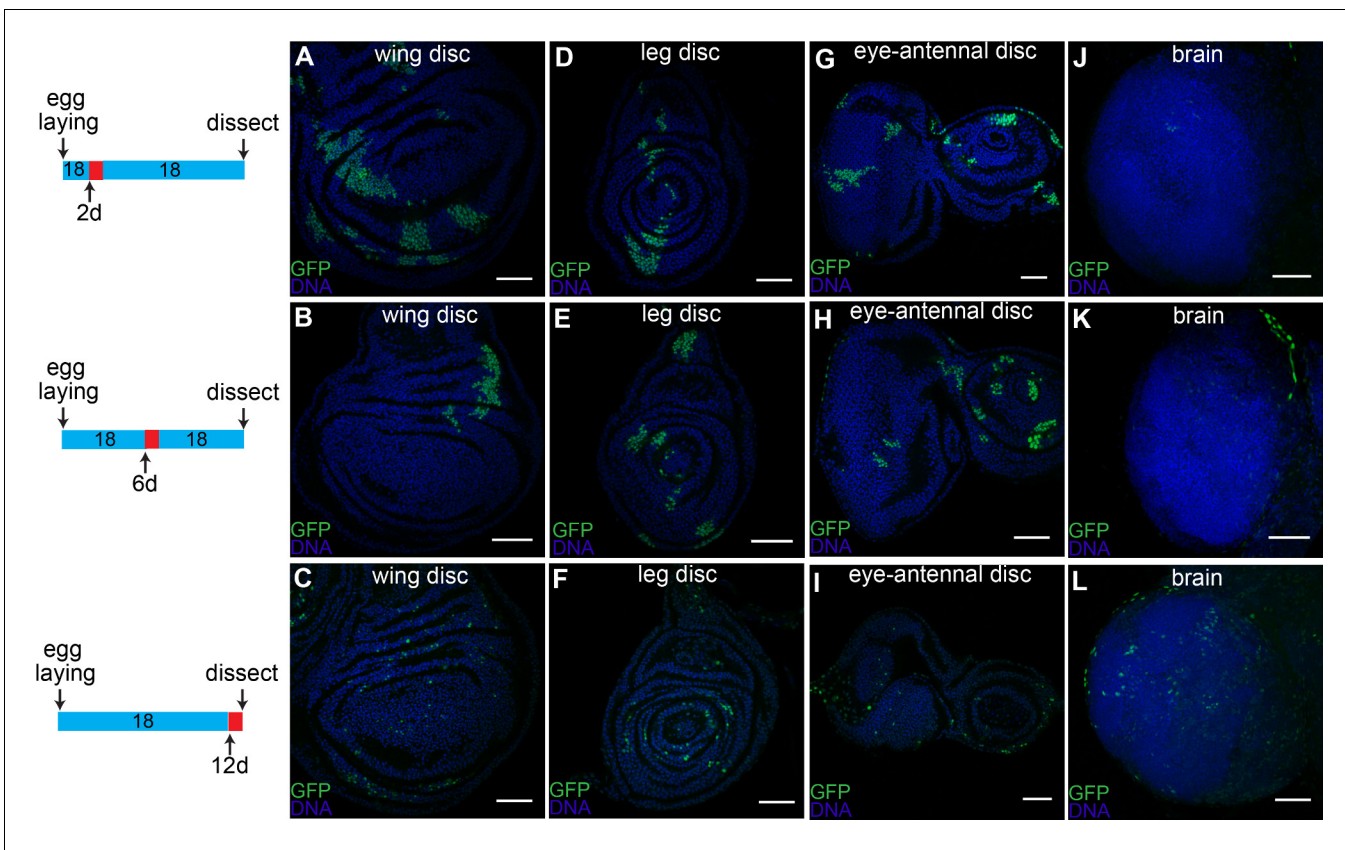

**Figure 5.** Timing of caspase activation in larval tissues. Larvae with CasExpress, G-trace and Gal80$^{ts}$ were grown at 18°C (blue in the timeline bars on the left) for 2d (A, D, G, J), 6d (B, E, H, K), or 12d (C, F, I, L), shifted to 29°C (red in the timeline bars) for 1d, then kept at 18°C until late third instar. Induction of GFP expression occurs in wing discs (A–C), leg discs (D–F) and eye-antennal discs (G–I) throughout the larval stage; whereas few cells in brain (J–L) survive caspase activation before third instar. Scale bars are 50 μm.

The following figure supplement is available for figure 5:

**Figure supplement 1.** GFP, RFP and DAPI fluorescence in imaginal discs from flies carrying CasExpress, Gal80$^{ts}$, and G-TRACE raised at 18°C.

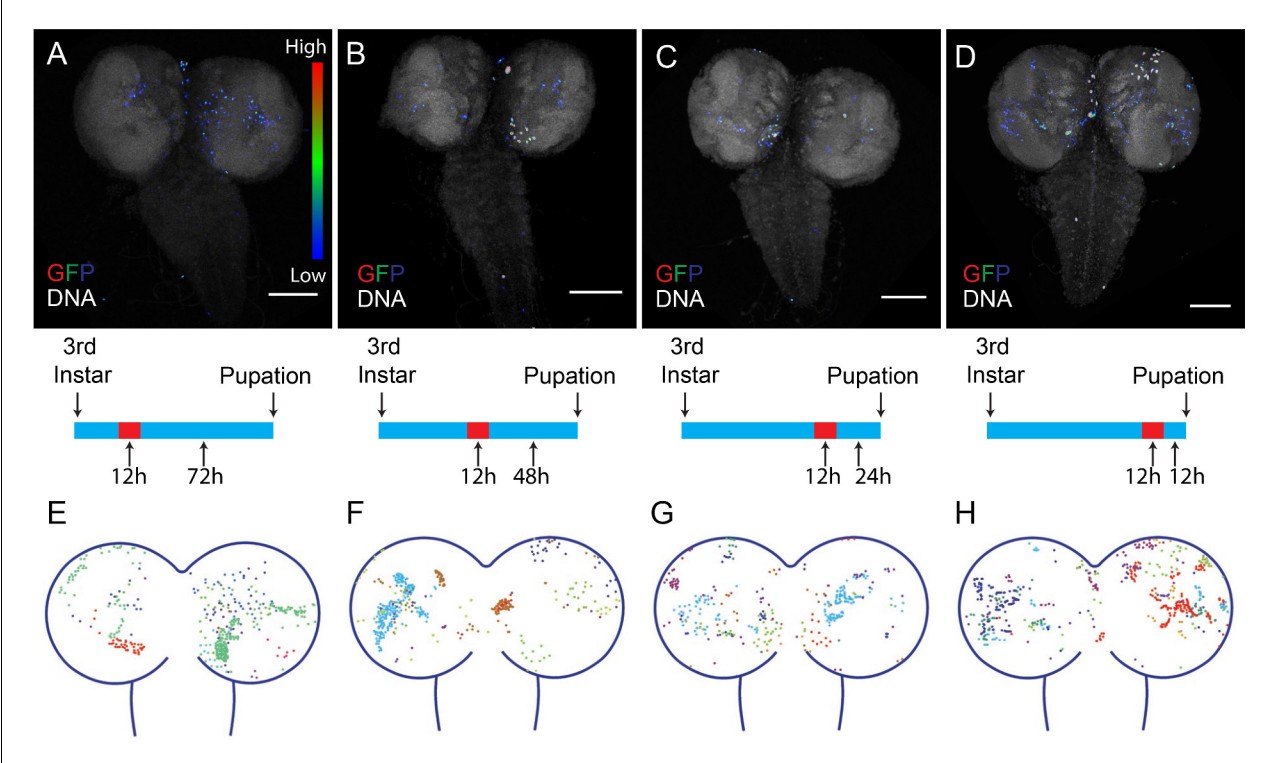

**Figure 6.** Timing of CasExpress activation in larval CNS. Larvae with CasExpress G-Trace and Gal80[ts] were grown in 18°C (blue bars in the middle) until 3[rd] instar and shifted to 29°C for 12 hr. Then larvae were kept at 18°C for 72 hr (**A, E**), 48 hr (**B, F**), 24 hr (**C, G**) or 12 hr (**D, H**) until they reached late 3[rd] instar. (**A–D**) Four examples of GFP expression patterns in the larval CNS presented in Rainbow RGB, which shows different levels of GFP intensity in different colors. (**E–H**) Z-projections of different samples were slightly transformed and fit into the diagram of brain. The positions of GFP positive cells for each sample are indicated with different colors. Scale bars are 100 μm.

The following figure supplement is available for figure 6:

**Figure supplement 1.** Caspase activity during larval CNS development.

## Imaginal disc CasExpress arises sporadically in time

The large percentage of GFP+ cells in late third instar larval tissues raised the question as to whether caspase was activated in a significant fraction of cells at one particular stage in development, or alternatively whether caspase was activated sporadically in time. To address this question, we grew Gal80[ts]/CasExpress/G-Trace flies at 18°C and then shifted them to 29°C for 24 hr either at the first instar (*Figure 5*, upper panels), the second instar (*Figure 5*, middle panels) or the mid third instar (*Figure 5*, lower panels). We then returned them to 18°C and dissected them at the late third instar larval stage. Rather than all the GFP+ cells arising at one particular stage, sporadic expression was observed regardless of when the temperature shift occurred. This was true in wing (*Figure 5A–C*), leg (*Figure 5D–F*) and eye-antennal discs (*Figure 5G–I*). Cells that activated the sensor later produced smaller patches of cells, as expected if the patches represent clonal descendants of a single event. However we cannot rule out the possibility that separated cells that activate the sensor, coalesced into patches based on differential adhesion. In contrast to the discs, caspase activation in the brain was limited to late larval stages (*Figure 5J–L*).

## The temporal and spatial pattern of brain CasExpress

Apoptosis plays a particularly important role in the nervous system, and cleaved caspase is detected throughout larval CNS (CNS) development, both in w[1118] and in DQVD sensor flies (*Figure 6—figure supplement 1*). Therefore we characterized the temporal and spatial activation of CasExpress in this tissue in more detail. We combined CasExpress, G-Trace and Gal80[ts]. Flies kept at 18°C to

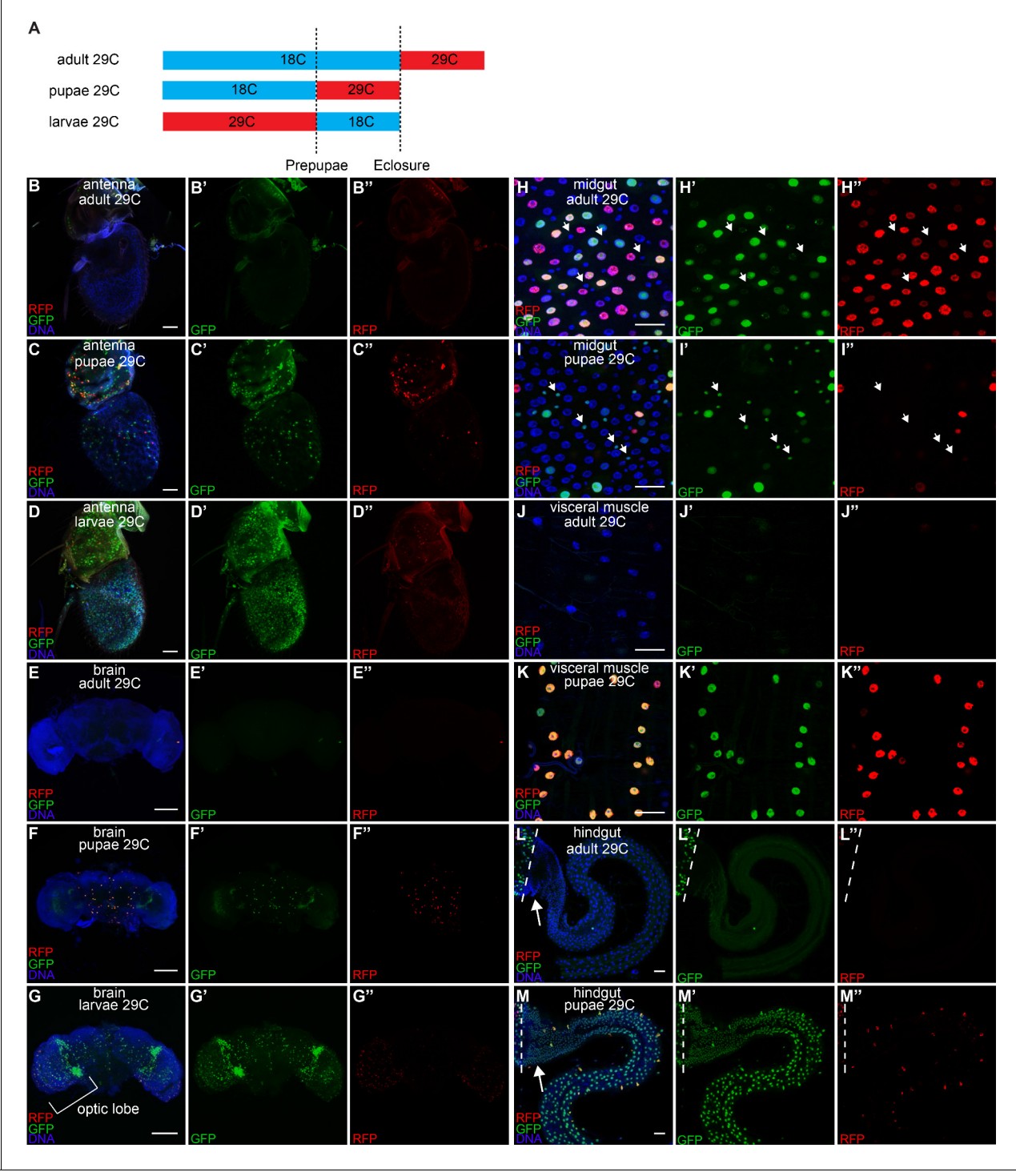

**Figure 7.** Developmental timing of caspase activation in adult tissues. (**A**) A schematic of the timing of the temperature shifts (blue: 18°C, red: 29°C) during the growth of flies with CasExpress, G-trace, and Gal80$^{ts}$. (**B–M**) GFP and RFP expression in antenna (**B–D**), brain (**E–G**), midgut (**H–I**), visceral muscle surrounding midgut (**J–K**), and hindgut (**L–M**) in flies with CasExpress, G-trace, and Gal80$^{ts}$ that grown at the condition indicated in the panels. Panels marked with prime showed separated channels of the left. Arrows in **H–I''** point to some examples of GFP+ progenitor cells. Dotted lines in **L–M''** mark the boundary between midgut and hindgut. Arrows in L and M point to the hindgut proliferation zone. Scale bars in **B–D**, and **H–M** are 25 μm. Scale bars in **E–G** are 100 μm.

silence CasExpress throughout development exhibited virtually no expression of RFP or GFP. However if we shifted them to 29°C for 12 hr at the early (*Figure 6A*), middle (*Figure 6B*), or late (*Figure 6C,D*) third instar and dissected them near the end of larval development, we observed GFP expression in seemingly random locations. Similar numbers of GFP-expressing cells appeared regardless of precise developmental stage. *Figure 6E–H* shows the patterns observed in 10 different animals, each in a different color, demonstrating the variability. The patterns were clearly not bilaterally symmetrical. We conclude that cells survive caspase activation sporadically during CNS development. This pattern seems more consistent with that expected for developmental anastasis, that is recovery from the brink of apoptotic cell death, rather than a precise role for caspase in the development of a specific cell type.

## CasExpress activation during metamorphosis

Metamorphosis requires remodeling of some tissues and wholesale destruction and rebuilding of others (*Baehrecke, 2002*; *Yu and Schuldiner, 2014*). Apoptosis contributes substantially to these processes. To address how much of the adult expression arose during metamorphosis in each tissue, we crossed in the Gal80$^{ts}$ repressor and grew flies at 18°C to prevent the induction of CasExpress. We then shifted the flies to 29°C, the non-permissive temperature for Gal80$^{ts}$, to allow induction only during specific time windows corresponding to larval, pupal, or adult stages respectively (*Figure 7A*). Distinct patterns were observed in different tissues. In the antenna (*Figure 7B–D"*) and brain (*Figure 7E–G"*), CasExpress was activated during larval and pupal stages but virtually none was detected in adulthood. Activation during the pupal period could be responsible for remodeling of connections during metamorphosis and was not unexpected, however the more extensive activity during the larval period suggests an additional function for caspase in earlier nervous system development. In midgut enterocytes, some activation occurred during pupal life but more appeared in the adult (*Figure 7H–I"*), possibly related to the biology of midgut enterocytes which face damage and undergo rapid turnover in adults even under normal physiological conditions. In visceral muscle surrounding the midgut (*Figure 7J–K"*) and in the hindgut (*Figure 7L–M"*) activation was limited to pupal stages, consistent with a role for caspase in metamorphosis of this tissue.

When CasExpress induction was allowed only during the pupal stage, some progenitor cells in the midgut (*Figure 7I–I''*), visceral muscle surrounding it (*Figure 7K–K''*), and the whole hindgut including the proliferation zone, which contains progenitor cells (*Figure 7M–M''*), showed GFP expression. Thus caspase was activated during metamorphosis and some cells survived. This is intriguing because during metamorphosis the larval gut degenerates and the adult gut is reconstituted by progenitor cells (*Micchelli, 2012*). We did not detect RFP in progenitor cells at any stage that we analyzed, and we only detected GFP in progenitors when CasExpress was allowed to be active during the pupal stage. Therefore caspase is likely activated for a brief period during pupal life. The progenitor cells, like the rest of the animal, are exposed to apoptotic stimuli such as systemic ecdysone (*Jiang et al., 1997*), yet they survive to reconstitute the adult gut. They might survive either because they are particularly resistant to caspase activity and apoptosis, as is postulated for stem/progenitor cells generally. Alternatively caspases may actually promote their proliferation or maintenance as has been described for some mammalian progenitors (*Li et al., 2010*; *Yoneyama et al., 2014*); or some progenitor cells may undergo anastasis and recover from the brink of apoptotic cell death. Although we cannot currently distinguish definitively between these possibilities, the observation that a subset of progenitor cells activates CasExpress might indicate that some cells resist the apoptotic stimulus prior to activation of caspase-3 whereas others experience caspase activity and recover from it. The observation that pupal visceral muscle cells exhibit RFP and GFP in nearly every cell suggests prolonged caspase activation. Together these observations demonstrate that survival of caspase activation occurs in distinct spatial and temporal patterns for different cell types and tissues, possibly due to differing epigenetic states, developmental mechanisms, and/or physiological functions (see Discussion).

## Functional significance of developmental caspase in the wing

We wondered if the observed caspase activity was functionally significant. The homozygous *Drice* mutant is lethal, as are *dronc* mutants. The interpretation is that these mutations prevent apoptosis, and that apoptosis is essential. Our results suggest an additional possibility, which is that non-

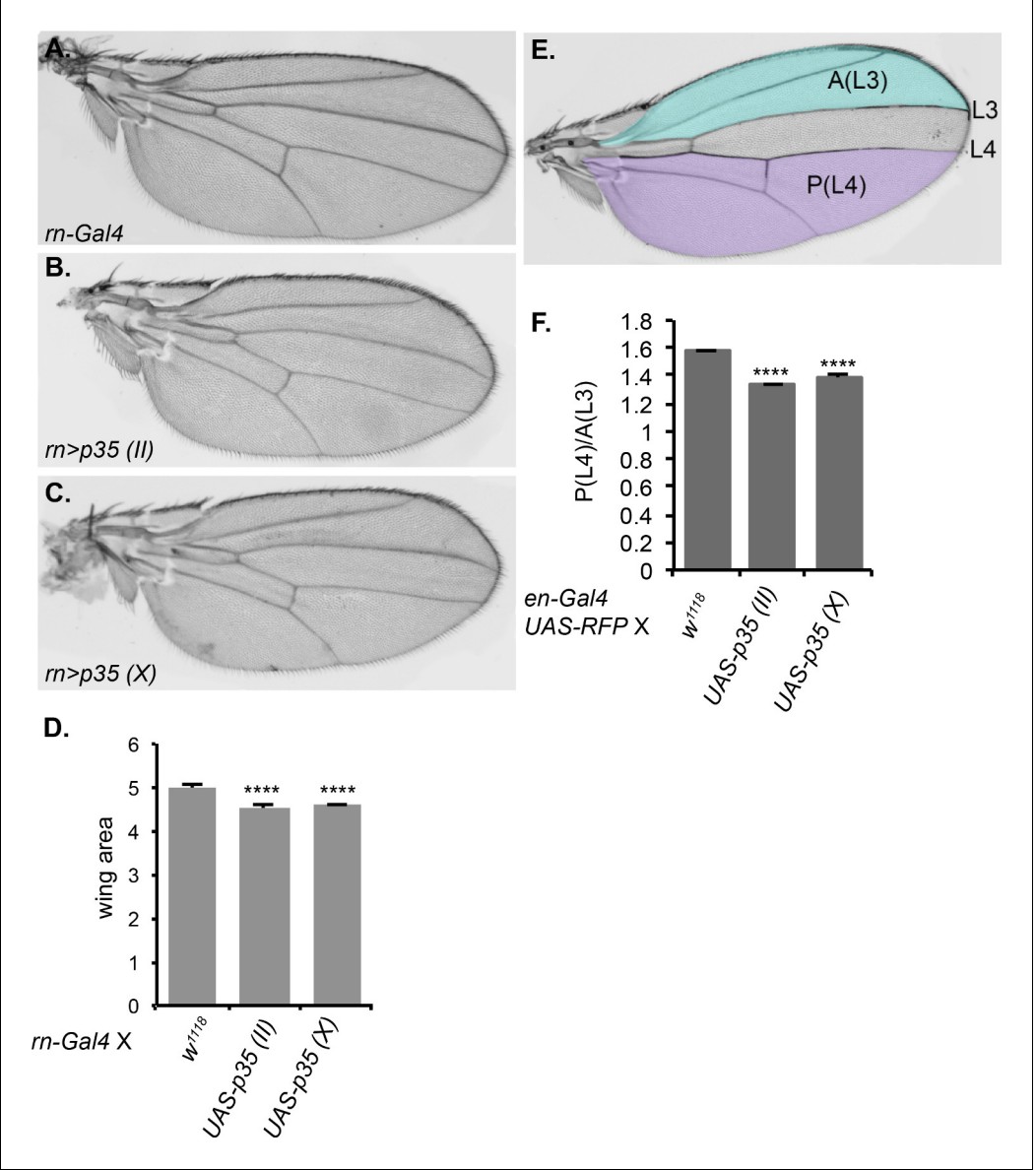

**Figure 8.** Inhibition of caspase activity reduces wing size. (**A–C**) Representative wings from progeny of rn-Gal4 crossed to (**A**) control w[1118], (**B**) UAS-p35 on chromosome II, or (**C**) UAS-p35 on the X chromosome. (**D**) Quantification of wing area in arbitrary units. (**E**) Schematic showing the regions used for area measurement in wings with or without p35 expressed under en-Gal4. In the anterior compartment, we measured the area anterior to L3 vein, which is highlighted in blue and marked as A(L3). In posterior compartment, we measured the area posterior to L4, which is highlighted in purple and marked as P(L4). (**F**) Quantification of the ratio between P(L4) and A(L3) in wings from progeny of en-Gal4 crossed to w[1118], UAS-p35 on the second chromosome, and UAS-p35 on the X chromosome. Error bars show standard error of the mean, and **** indicates $p<0.0001$.

apoptotic caspase activity may be important during normal development. To test this, we crossed two different UAS-p35 transgenes to the *rotund-Gal4* (rn-Gal4) line, which expresses in the pouch region of the wing imaginal disc, the region that gives rise to the adult wing blade. We then evaluated the morphology and size of the adult wing for defects in growth and/or patterning. Although the wings appeared normally patterned, they showed a small (10%) but reproducible and significant reduction in area (*Figure 8A–D*), demonstrating the functional importance of caspase activity in this tissue. We repeated this experiment using engrailed-Gal4, which drives expression only in the posterior compartment of the wing, and compared the area of the posterior compartment to that of the

anterior compartment as an internal control. Again, inhibition of apoptosis by expression of p35 caused a small but significant reduction in size. If the only function of caspase were to promote apoptosis, inhibition of caspase should result in excess cells, and therefore a larger size. The observation of a smaller wing suggests a different function for caspase in this tissue (see Discussion).

## Discussion

Here we report the first systematic analysis of the fates of cells that survive caspase-3 activation throughout Drosophila development. The striking results include widespread cell survival of caspase activation and the distinct spatial and temporal patterns observed in different tissues throughout development. Caspase-3 activation has been strongly associated with cell death (*Thornberry, 1998*; *Chipuk et al., 2006*). While some previous studies have indicated that caspase-3 can perform other functions (e.g.[*Connolly et al., 2014*]), these non-apoptotic caspase activities are generally considered exceptions to the rule. However it has not been possible to systematically follow the fates of cells that experience executioner caspase activity throughout development in vivo.

We observed several distinct patterns of CasExpress activation, which likely reflect different biological functions. Oenocytes and cells of the salivary gland and Malpighian tubules activated CasExpress in every cell, in every animal, with no evidence of apoptosis or even partial cellular destruction. This pattern seems most consistent with non-apoptotic roles for caspases. One known protein target of Drice and Dcp1 that might be relevant in this context is the Sterol Regulatory Element-binding Protein (dSREBP) (*Amarneh et al., 2009*). SREBP is synthesized as a membrane-tethered precursor that is released by proteolytic cleavage so that it can translocate to the nucleus where it transcribes target genes involved in lipid synthesis and uptake. Oenocytes have an established role in lipid synthesis (*Makki et al., 2014*). According to FlyBase, SREBP is expressed at high levels in many of the tissues that show constitutive caspase-3 activity. SREBPs in mammals and flies are cleaved by site-2 protease (S2P). However Drice and Dcp1 can also cleave SREBP and can even substitute in the absence of S2P in Drosophila (*Amarneh et al., 2009*). It is possible then that one function of caspase activity in oenocytes and other cell types is to activate SREBP during normal development and/or in times of stress to meet metabolic demands.

The published literature provides clear examples of caspase activities that promote cellular remodeling via limited destruction such as sperm maturation (*Arama et al., 2003*; *Huh et al., 2004*), remodeling of neurites (*Yan et al., 2001*; *Finckbone et al., 2009*), and enucleation of certain terminally differentiating cells, such as erythrocytes and lens epithelial cells (*Connolly et al., 2014*). In such tissues one would also expect to see CasExpress activated in a reproducible temporal and spatial pattern. Metamorphosis is a period of insect development during which extensive tissue remodeling occurs. The CasExpress activity that we detected during pupal life in the nervous system for example could be a consequence of remodeling.

In contrast, in imaginal discs and the brain CasExpress was activated sporadically in both space and time, during periods of development when apoptosis is known to occur. This pattern seems most consistent with developmental anastasis. Anastasis was first described in cultured cells and is defined as the recovery of cells from the brink of apoptotic cell death after caspase-3 activation (*Tang et al., 2012*). During normal development of many tissues including Drosophila imaginal discs and mammalian blastocysts, cells are thought to compete for survival based on differential fitness (*Moreno and Rhiner, 2014*; *Merino et al., 2015*; *Kolahgar et al., 2015*; *Vincent et al., 2013*; *de Beco et al., 2012*). Differences in fitness can be detected amongst cells with artificially induced differences in growth rates, caused by differential ribosomal protein levels (*Morata and Ripoll, 1975*), c-Myc expression levels (*de la Cova et al., 2004*), or access to trophic factors (*de Beco et al., 2012*). In the nervous system for example, those cells that obtain sufficient growth factor signaling survive and those that receive too little undergo apoptosis. This has generally been considered an all-or-nothing decision. However the observation that cultured cells exposed transiently to a lethal toxic stimulus can recover after caspase-3 activation, or survival of cells exposed to a sublethal dose of radiation or mitochondrial permeabilization (*Liu et al., 2015*; *Ichim et al., 2015*), raise the question as to whether cells might actually bounce back from transient caspase activation during development as well. Our results show that cells can survive caspase-3 activation during normal development, perhaps due to recovery from a transient apoptotic stimulus.

Another possibility is that cells in a population differ in their sensitivities to apoptosis due to variation in epigenetic states (*Flusberg and Sorger, 2015*; *Spencer and Sorger, 2011*). In mammals the E3 ubiquitin ligase PARC can target cytoplasmic cytochrome c for ubiquitin-mediated degradation, providing one molecular mechanism by which cells can recover from an apoptotic stimulus (*Gama et al., 2014*). Differential expression of PARC may confer different levels of resistance to executioner caspase activity. Other, as yet unknown, epigenetic differences between cells may also confer differing sensitivities to caspase activation.

Our results suggest that the ability of cells to survive caspase activation changes during development. Many cells in the embryo activate caspase-3, yet we detected no GFP expression in embryos, except in the salivary gland. Our temperature shift experiments revealed that, in imaginal discs for example, more and more cells survived as development progressed, and cells that activated the sensor in early stages produced large numbers of progeny such that by the end of the third larval instar, the majority of cells expressed GFP.

A number of studies report immunoreactivity against cleaved caspase-3 in neurons that appear to be dividing, differentiating or migrating (*Yu and Schuldiner, 2014*; *Yan et al., 2001*; *Finckbone et al., 2009*; *Schoenmann et al., 2010*). While tantalizing, the studies were carried out in fixed tissue so the ultimate fates of such cells could not be determined. In the current study we were able to follow the fates of cells that survived caspase activation, and these results demonstrate that in many tissues of the adult the majority of cells arise from cells that experience transient caspase activity at some point during their development. Therefore such events are not the exception; rather they are the rule.

## Autonomous versus non-autonomous survival

When extra apoptosis is artificially induced in Drosophila imaginal disc cells, it stimulates surviving cells to proliferate (*Fan and Bergmann, 2008*). Dying cells secrete growth factors to facilitate the survival and proliferation of their neighbors in the process known as compensatory cell proliferation (*Xing et al., 2015*; *Kuranaga et al., 2011*). The marking system that we report here demonstrates cell autonomous survival of caspase activation. Both autonomous and non-autonomous survival and proliferation may cooperate to promote recovery of tissues from insults that kill some but not all cells.

A role for caspases in injury repair and tissue regeneration has been demonstrated in Hydra, Xenopus, planaria, newts and in mouse liver (reviewed in [*Connolly et al., 2014*]), indicating that this is a well-conserved and general phenomenon. Our observation that the majority of cells in the adult fly descend from cells that survive caspase activation at some point suggests that, in addition to the well-documented compensatory proliferation in response to injury, there may autonomous compensatory proliferation in cells that survive caspase-3 activation during normal development. The idea is that some cells die and need to be replaced so the cells that survive proliferate. Such an autonomous increase in proliferation might explain the abundance of GFP-expressing cells in the adult. It could also explain the otherwise paradoxical result that inhibition of executioner caspase activity in the wing imaginal disc by p35 reduced wing area in the adult. If inhibiting caspases only blocked apoptosis, one would expect the tissue to contain excess cells and to be either larger, abnormally patterned, or both. In contrast we observed a small decrease in wing area, consistent with the idea that inhibiting caspase activity might also inhibit compensatory cell proliferation during normal development. An earlier study (*de la Cova et al., 2004*) showed that inhibiting apoptosis in the wing disc led to variability in the size of the disc later in development; however this study did not address the ultimate effect on the size of the adult wing. It will be interesting in the future to examine CasExpress in models of injury, repair and regeneration to determine if cell autonomous compensatory proliferation occurs in those settings as well.

## Additional examples of developmental anastasis

Two papers document examples of cell recovery from apoptosis during *C. elegans* development (*Reddien et al., 2001*; *Hoeppner et al., 2001*). When phagocytosis is impaired, a fraction of cells that normally die are able to reverse the morphological signs of apoptosis, which are caused by caspase-3 activity. These cells not only survive, they differentiate. One interpretation of these findings is that phagocytosis normally occurs so early in the death process that it prevents anastasis. However

development in *C. elegans* is far more stereotyped than it is in most organisms. In *C. elegans* the fate of every single cell is precisely determined. However in organisms with greater numbers of cells, cell survival or death is not thought to be a predetermined cell fate; rather there is a selection process in which cells compete (*Moreno and Rhiner, 2014*; *Merino et al., 2015*; *Vincent et al., 2013*; *de Beco et al., 2012*). Our results indicate that many cells in adult flies derive from cells that survive caspase activity at some point during their development. An alternative interpretation of the *C. elegans* studies is that the ability to survive and recover even after caspase-3 activation is a fundamental and ancient cellular property that evolved early and still exists in a latent form, even in an animal that does not normally need it. Even in *C. elegans*, the precise moment of engulfment is not predetermined; and it is not always the same cell that consumes the dying cell. Therefore in organisms with larger numbers of cells whose fates are far less predictable, it is unlikely that engulfment always occurs at a precise time point during the apoptotic process. The results presented here demonstrate that it is not rare for cells to survive caspase-3 activation during normal Drosophila development, and such cells make a major contribution to normal adult tissues.

## Materials and methods

### Fly strains

The following transgenic and mutant strains were used:

The CasExpress biosensor (pattB-Ubi-CasExpress-DQV<u>D</u>) and caspase-insensitive control (pattB-Ubi-CasExpress-DQV<u>A</u>) were newly generated as follows. First, a backbone pattB-synaptobrevin-7-QFBDAD-hsp70 (gift from Christopher J. Potter lab) was linearized with restriction enzymes AatII and BamHI. The poly-ubiquitin promoter was cloned by PCR from pUWR (Addgene, Cambridge, MA), and the product was inserted to backbone by In-Fusion Cloning Kit (Clontech Laboratories, Mountain View, CA). The product, which was verified by sequencing was named pattB-Ubi. Second, pattB-Ubi was linearized with restriction enzymes NdeI and PstI as a backbone. An insert consisting of the sequence of MCD8, DIAP1 (residues 2–147) and Gal4 in 5' to 3' order and two 15 bp sequences overlap with backbone on both 3' and 5' end was generated by PCR and In-Fusion cloning. Residues 21 and 22, immediately following the DQVD cleavage site in DIAP1, were mutated from sequence NN to GV, in order to protect the cleaved product from possible N-end rule degradation. Third, the insert and backbone were ligated using the In-Fusion kit. A product verified by sequencing was named pattB-Ubi-CasExpress-DQVDGV. Finally, nucleotide 59 of DIAP1 sequence in pattB-Ubi-CasExpress was mutated to change amino acid 20 from D to A by single point mutagenesis. A product verified by sequencing was named pattB-Ubi-CasExpress-DQVAGV. CasExpress and control plasmids were sent to BestGene Inc., inserted to Perrimon strain P{CaryP}attP40 through a PhiC31 integrase mediated transgenesis. Random insertions of CasExpress-DQVDNN and CasExpress-DQVANN were also generated. Dronc[I29] was a gift from Kenneth D. Irvine. The following strains were obtained from the Bloomington Stock Center: G-Trace (Bloomington #28280); tub-Gal80[ts] (Bloomington #7018); GMR-p35 (Bloomington #5774); UAS-p35 BH1 and BH2 (Bloomington #5072 and 5073). All lines and crosses were kept at 25°C except otherwise indicated.

### Dissection, immunohistochemistry and imaging

Larval intestines, oenocytes (together with surrounding cuticle) and adult muscles, brains, eyes, ovaries, oviducts, uteri, tissues were dissected in PBS. For adult ventral nerve cords, whole thoraxes were used for fixation. For larval tissues, the anterior 1/3 part of larvae was cut off and turned inside out, all tissues remained attached to cuticle during fixation. Tissues were fixed in 4% paraformaldehyde in PBS at room temperature for 10 min (larval cuticles with CNS and imaginal discs) 30 min (adult thoraces). Other tissues were fixed for 15 min. After fixation, adult ventral nerve cords were dissected from adult thoraces. The samples were then washed with PBS/0.3% Triton X-100 (PBSt) for 3 x 10 min and blocked with 5% goat serum for 30 min. Fluorescence of RFP and GFP were detected directly without antibody staining. Mouse anti-mCD8 (Santa Cruz, Dallas, TX, #51735, 1:50), and rabbit anti-Cleaved Dcp-1(Asp216) (Cell Signaling, Danvers, MA, #9578, 1:100) were incubated with dissected tissues overnight at 4°C, followed by 3 x 10 min PBSt washing and secondary antibody incubation for 2 hr at room temperature. Samples then were washed twice for 15 min each with

PBSt and incubated for 15 min with 10 ng/ml Hoechst 33,342 in PBSt. After Hoechst staining, larval CNS and imaginal discs were dissected away from cuticle. All samples were mounted in Vectashield mounting media (Vector Laboratories, Burlingame, CA, H-1000).

A Zeiss AxioZoom microscope was used for imaging whole larvae. A Zeiss LSM 780 confocal microscope was used for the rest of images. Embryo collections, fixation and acridine orange (Sigma-Aldrich, St. Louis, MO, A6014) staining of embryos are as described (*McCall and Peterson, 2004*)

### Image analysis and quantification

Images were processed with Fiji. A threshold for each channel of interest (COI, e.g. RFP, GFP) was set by auto-threshold (method: Default, Dark). For anti-cDcp1 the threshold was set using MaxEntropy. Area above threshold was measured as S[COI]. The area of DNA was measure in the same manner as S[DNA]. A ratio of S[COI]/S[DNA] was then calculated.

### GMR-p35 suppression of CasExpress in antennal-eye disc

To test suppression of CasExpress by the caspae inhibitor p35, larvae with the genotype GMR-p35; CasExpress/G-Trace; TubGal80$^{ts}$ were raised in 18°C until early third instar. Larvae were then transferred to 29°C and incubated for 48 hr. Antennal-eye discs were then dissected, fixed and stained with Hoechst 33342, followed by a Z-stack imaging on LSM780 microscope (Objective: 20x Zeiss plan-apochromat dry, 0.8 NA; step-size: 1.46 μm).

Images were process with Fiji. A Z-projection of each image was generated by maximum intensity algorithm. An ROI was drawn to define the boundary of the antennal disc. A threshold for the RFP channel was determined by auto-threshold (method: Default, Dark). Another ROI was then drawn to define the boundary of region of the eye disc posterior to the morphogenetic furrow. Threshold of RFP channel of original image without Z-projection was set as (0.5a, b). Area of RFP above threshold was measured for each layer and summed. Area of DNA above threshold determined by auto-threshold (method: Default, Dark) was also measured for each layer and summed. The ratio of summed areas of RFP and DNA was then calculated.

### Wing size quantification

Crosses were maintained at 25°C. The progeny of the desired genotypes were collected and dehydrated in 100% ethanol. The wings were then mounted in Canada balsam (Gary's magic mountant, Sigma) and photographed using a Zeiss AxioZoom microscope. Wing sizes were quantified using ImageJ software.

### Statistical analysis

Statistical significance was determined using the unpaired two-tailed t test for two-sample comparison or one-way ANOVA for multiple samples analysis, with $p < 0.05$ set as the threshold for significance. The Tukey test was used to derive adjusted P values for multiple comparisons. In figures, *** indicates $p < 0.001$, and **** indicates $p < 0.0001$. Error bars show standard error of the mean.

## Acknowledgements

This work was supported by NIH grants R01GM46425 and R21EY022498. We thank Thomas Sun for technical assistance and Ugochukwu Ihenacho for useful discussions. H L Tang generated the random insertions of CasExpress-DQVDNN and CasExpress-DQVANN during his 2010 - 2014 tenure in the lab, with assistance from H M Tang.

## Additional information

### Funding

| Funder | Grant reference number | Author |
| --- | --- | --- |
| National Eye Institute | R21EY022498 | Denise J Montell |

| National Institute of General Medical Sciences | R01GM46425 | Denise J Montell |

The funders had no role in study design, data collection and interpretation, or the decision to submit the work for publication.

### Author contributions

XD, GS, YGA, Conception and design, Acquisition of data, Analysis and interpretation of data, Drafting or revising the article; JOW, Acquisition of data, Analysis and interpretation of data, Drafting or revising the article; SE, Acquisition of data, Analysis and interpretation of data; DJM, Conception and design, Analysis and interpretation of data, Drafting or revising the article

### Author ORCIDs

Denise J Montell, http://orcid.org/0000-0001-8924-5925

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
