## [Decision Letter]

[Editors’ note: this article was originally rejected after discussions between the reviewers, but the authors were invited to resubmit after an appeal against the decision.]

Thank you for submitting your work entitled "Widespread cell survival following caspase-3 activation in vivo" for consideration by *eLife*. Your article has been reviewed by two peer reviewers, and the evaluation has been overseen by a Reviewing Editor and Fiona Watt as the Senior Editor. Our decision has been reached after consultation between the reviewers. Based on these discussions and the individual reviews below, we regret to inform you that your work will not be considered further for publication in *eLife*.

*Reviewer #1:*

In this interesting paper, the authors report work in which they built a tethered Gal 4-based caspase sensor that reports caspase activation during *Drosophila* development, in real time and, importantly, also that activated in the past, meaning that cells may have survived caspase activation (which the authors call "anastasis"). Cleavage of a caspase sequence inserted in between a plasma membrane tether (CD8) and Gal 4 releases Gal 4 to the nucleus where it activates UAS-RFP. A flp-out "memory" cassette (now known as G-trace) allows heritable GFP expression in the progeny of a cells that has activated the sensor. Cells that are both RFP and GFP positive are those that have activated the sensor in the past and also continue to activate it. Those with only GFP are those that derived from a previous activation event.

The authors observe sensor activation in numerous tissues in the fly at different developmental stages. Using a Gal80ts to inhibit Gal 4 activity at selected times in development they demonstrate interesting differences between tissues in sensor activation, suggesting that tissues activate caspases according to differences between tissue physiologies and developmental stages. These are very interesting observations that suggest that caspases are activated at numerous points in development and that this does not necessarily lead to death of the cell, at least not immediately. Although it is known that many cells die either sporadically or in response to programmed cues these data suggest that it may occur more frequently than previously thought. The validation of the sensor via a variety of controls seems solid, and overall they present a very nice tool that should be helpful for exploring in more detail the role of caspases during development. However, the authors make several conclusions that I think are less convincing, and thus reduces my enthusiasm.

For example, I'm not sure that it's fair to say that the GFP+ cells are all survivors of caspase activation. It is more likely that one of them is, which has then given rise to all of the other cells in the patch. Alternatively, even the parent cell has died but not before producing progeny that will inherit the flipped-out cassette allowing expression of GFP. It is also not clear whether the GFP+ (and RFP+ cells, for that matter) are truly "clones" or whether they sort together due to adhesive alterations due to caspase activation.

The authors test for a physiological function of caspase activation by blocking its effects through expression of the pan-caspase inhibitor, p35. However, it should be noted that the loss of size control from blocking death in discs with expression of p35 has been previously demonstrated (de la Cova et al., Cell 2004).

In the Discussion the authors bring up the idea that the sensor activation in imaginal discs, which occurs at several different points during larval development, could be due to competition between imaginal disc cells for growth factors. There is no citation for this supposition, and it would be helpful if they provided one. Although such competition clearly occurs in the brain, it is not at all clear this occurs in discs. The idea was previously proposed to occur (Moreno et al., Nature 2002) but the data remain controversial (later the same lab was unable to reproduce this effect; Martin et al., Development 2009). Another explanation the authors might consider is that is that genetic/epigenetic alterations arise in some cells that render them less sensitive to growth or survival factors and thus lead to caspase activation.

Finally, in reading this manuscript I was puzzled by the citation of a previous paper from earlier this year that reported very similar results using essentially an identical sensor. It became clear upon reading the Materials and methods section that the first authors of that paper (Tang et al., Sci Rep 2015) were previously in the Montell lab; it is cited here that the "design and development of the approach and most of the reports were first obtained in the Montell lab". This is quite unfortunate, but since this was already published it seriously reduces the novelty of the present work. The present work does go a bit farther beyond what was reported earlier, with targeted genomic insertion, a more extensive analysis of different tissues and timing, and the report of a wing phenotype (although as noted above this also has been demonstrated previously).

*Reviewer #2:*

In this manuscript, the authors describe a method for detecting current or past caspase activity in *Drosophila*. The CasExpress transgene is based on the GTrace method, and labels cells that are experiencing current caspase activity with RFP, and cells that have experienced past caspase activity and their progeny with GFP. A previously published paper describes a very similar construct (CaspaseTracker), but this work adds additional description and controls.

The most interesting aspect of the paper is the very large number of cells that have experienced caspase activation in the past or in progenitors. However, it is somewhat specious to claim that it is a surprise that caspase activity doesn't always kill cells. In fact, the first caspase (Caspase1) is not mainly a proapoptotic caspase. The caspase dredd in flies mainly plays a role in innate immunity. (In fact, if dredd can cleave the construct, the widespread gut staining could be explained by the immune response). A number of examples of non-apoptotic functions for pro-apoptotic caspases are also well described. However, this tool will be helpful to identify potential additional non-apoptotic caspase functions.

There are some concerns however, starting with the CasExpress construct. In theMethods, the numbering for the DIAP segment inserted into the construct seems to be incorrect. Previous work (which should be cited) shows that the cleavage site is at amino acid 20 (and the N-terminal aa that were changed are 21 and 22). Furthermore, and more problematic, is that this construct contains the entire BIR1 of DIAP1, which has caspase inhibitory function (Tenev et al. '04). Although the membrane bound construct may not inhibit cell death, it would be important to show that the cleaved form is not inhibitory. This could explain some of the survival of cells that had experienced caspase activation.

To examine the origin of the widespread caspase activity reporter shown in the adult, the authors look at the reporter expression in the embryo. However, the data shown in Figure 2 is somewhat puzzling. The embryo shown is almost certainly not stage 10 as stated, based on salivary gland morphology. It may be stage 14. At this stage, significant ongoing cell death, as detected with Acridine orange, TUNEL or cCP3, has been reported. Why is there so little RFP in the embryo? The Dronc null and p35 experiments are excellent controls showing that the reporter is caspase dependent, but does it somehow miss earlier caspase activation?

The wing size experiment shown in Figure 8 could suggest that caspase activity contributes in a positive way to normal wing size. It is important to show that other transgenes expressed from this driver do not suppress wing growth. The driver alone is not a sufficient control. Also the graph exaggerates the wing size difference by starting at 4.2 rather than 0. The wing size should be normalized to another tissue, to account for differences in body size.

[Editors’ note: what now follows is the decision letter after the authors submitted for further consideration.]

Thank you for resubmitting your work entitled "CasExpress reveals widespread and diverse patterns of cell survival of caspase-3 activation during development in vivo" for further consideration at *eLife*. Your revised article has been favourably evaluated by Fiona Watt (Senior editor), Utpal Banerjee (Reviewing editor), and two reviewers. The manuscript is much improved but there are some remaining issues that need to be addressed before acceptance, as outlined below:

The reviewers strongly feel, and you will likely agree, that it is critical to make sure that the caspase being activated in your system, and detected by the sensor is an apoptotic caspase rather than one related to immune response. Although the p35 experiment is valuable, the reviewers have pointed out that it will be important to show whole larval expression, particularly the gut which is quite active in its immune capability. Also, it is rather easy to demonstrate, by placing the sensor directly in a dredd mutant background, to rule out the possibility that this pathway might be involved. Thus, please include the image of the whole larva in p35 and also show sensor expression in a dredd mutant background (X-chromosome location of dredd makes this an easy cross to score in larvae).

---

## [Author Response]

[Editors’ note: the author responses to the first round of peer review follow.]

*Reviewer #1: The authors observe sensor activation in numerous tissues in the fly at different developmental stages. Using a Gal80ts to inhibit Gal 4 activity at selected times in development they demonstrate interesting differences between tissues in sensor activation, suggesting that tissues activate caspases according to differences between tissue physiologies and developmental stages. These are very interesting observations that suggest that caspases are activated at numerous points in development and that this does not necessarily lead to death of the cell, at least not immediately. Although it is known that many cells die either sporadically or in response to programmed cues these data suggest that it may occur more frequently than previously thought. The validation of the sensor via a variety of controls seems solid, and overall they present a very nice tool that should be helpful for exploring in more detail the role of caspases during development. However, the authors make several conclusions that I think are less convincing, and thus reduces my enthusiasm. For example, I'm not sure that it's fair to say that the GFP+ cells are all survivors of caspase activation. It is more likely that one of them is, which has then given rise to all of the other cells in the patch.*

We agree with the reviewer and did not intend to make any other claim. We acknowledge that GFP+ cells are the sum of all cells that survive sensor activation and their progeny. We have carefully considered the writing to make sure that this point is clear.

*Alternatively, even the parent cell has died but not before producing progeny that will inherit the flipped-out cassette allowing expression of GFP.*

We believe that it is highly unlikely that a cell could divide while dying. In our in vitro experiments on cultured mammalian cells, we have never seen even a single example of a cell doing this. We do observe cells going to the brink of death, recovering, and then dividing. Or they die. But we have never observed a cell begin to undergo apoptosis and then divide without full recovery so that one daughter dies and the other one lives. Recovery seems to be a pre-requisite for survival.

*It is also not clear whether the GFP+ (and RFP+ cells, for that matter) are truly "clones" or whether they sort together due to adhesive alterations due to caspase activation.* The transient temperature shift experiments suggest that the groups of cells are likely clones. This is because the sizes of the cell clusters are larger when more time elapses between induction and analysis, consistent with an expanding clone. However, the reviewer’s alternative interpretation, which is an interesting idea, could certainly be true in some cases. Therefore, we have mentioned it in the Results (subsection “The temporal and spatial pattern of brain CasExpress”).

*The authors test for a physiological function of caspase activation by blocking its effects through expression of the pan-caspase inhibitor, p35. However, it should be noted that the loss of size control from blocking death in discs with expression of p35 has been previously demonstrated (de la Cova et al., Cell 2004).* The reviewer is correct that de la Cova et al. inhibited caspase activity by expressing p35 in wing discs. However, they analyzed the effect on the size of the late 3^rd^ instar larval wing disc, whereas we measured the effect on the final size of the adult wing. Importantly the result we present here has not been published before because de la Cova et al. looked at a different endpoint. It is interesting that there are similarities and differences between the two sets of findings. Neither of us observed the “expected” result, which would have been a larger organ (wing disc or wing) due to the presence of excess cells due to inhibition of cell death. They observed insteadincreased variability in the size of the developing disc, and we report asmaller adult wing. Both results are consistent with the idea that cell death plays a role in regulating tissue size, but not in the simple way that one would expect, i.e. removal of excess cells. Our result raises the possibility of cell autonomous compensatory proliferation following survival of caspase 3 activation, a concept not raised in any previous study.

*In the Discussion the authors bring up the idea that the sensor activation in imaginal discs, which occurs at several different points during larval development, could be due to competition between imaginal disc cells for growth factors. There is no citation for this supposition, and it would be helpful if they provided one. Although such competition clearly occurs in the brain, it is not at all clear this occurs in discs. The idea was previously proposed to occur (Moreno et al., Nature 2002) but the data remain controversial (later the same lab was unable to reproduce this effect; Martin et al., Development 2009).*

We now cite a number of articles that refer to the idea that disc cells might compete for limiting growth factors (de Beco et al., 2012; Moreno and Rhiner, 2014; Vincent et al., 2013). The reviewer is correct that one particular growth factor, Dpp, does not appear to be limiting; however, the general idea that cells might compete for limiting growth factors or nutrients is still very much alive as a possibility. Wingless is another candidate growth factor, as would be circulating factors or even nutrients.

*Another explanation the authors might consider is that is that genetic/epigenetic alterations arise in some cells that render them less sensitive to growth or survival factors and thus lead to caspase activation.* We agree that this is an interesting possibility. There may be intrinsic differences in cells that make them more or less sensitive to caspase activity levels. We also refer to the work of Peter Sorger (Flusberg and Sorger, 2015; Spencer and Sorger, 2011) for example, who studies differential sensitivity of cells to apoptotic stimuli. It is worth noting that most investigators have focused primarily on the idea that cells may differ in their sensitivity to stimuli that occur upstream of caspase activation because the field has not appreciated before that there even could be differential sensitivities to caspase activity itself, an idea supported by the current work as well as other very recent publications (Gama et al., 2014; Ichim et al., 2015).

*Reviewer #2: The most interesting aspect of the paper is the very large number of cells that have experienced caspase activation in the past or in progenitors. However, it is somewhat specious to claim that it is a surprise that caspase activity doesn't always kill cells. In fact, the first caspase (Caspase1) is not mainly a proapoptotic caspase. The caspase dredd in flies mainly plays a role in innate immunity. (In fact, if dredd can cleave the construct, the widespread gut staining could be explained by the immune response). A number of examples of non-apoptotic functions for pro-apoptotic caspases are also well described.*

We acknowledge those prior observations, but the extent of survival and the stage- and tissue-specific patterns are quite remarkable and unexpected.

*There are some concerns however, starting with the CasExpress construct. In theMethods, the numbering for the DIAP segment inserted into the construct seems to be incorrect. Previous work (which should be cited) shows that the cleavage site is at amino acid 20 (and the N-terminal aa that were changed are 21 and 22).*

We apologize for the confusion. The mutation was made in nucleotide 59 (we previously wrote “residue” instead of “nucleotide”), and it altered amino acid 20 and the other two N-terminal amino acids that were changed were in fact 21 and 22. We have also added the citation.

*Furthermore, and more problematic, is that this construct contains the entire BIR1 of DIAP1, which has caspase inhibitory function (Tenev et al.'04). Although the membrane bound construct may not inhibit cell death, it would be important to show that the cleaved form is not inhibitory. This could explain some of the survival of cells that had experienced caspase activation.*

In Figure 6—figure supplement 1 we showed that it is not inhibitory in the larval brain because cleaved Dcp1 levels were indistinguishable between *w1118* and sensor flies. The sensor is not expressed at high levels and is tethered to the membrane, providing an explanation for why it is not inhibitory. It also causes no discernible phenotype in any tissue and the animals are alive, fertile, and healthy. To make this point clear earlier in the paper, we have added text (subsection “Distinct spatial and temporal patterns of CasExpress during development”, second paragraph). We also added new data showing the very similar levels of acridine orange staining in *w1118* embryos compared to the DQVD and DQVA transgenic lines. This is now shown in Figure 2”.

*To examine the origin of the widespread caspase activity reporter shown in the adult, the authors look at the reporter expression in the embryo. However, the data shown in Figure 2 is somewhat puzzling. The embryo shown is almost certainly not stage 10 as stated, based on salivary gland morphology. It may be stage 14.*

The reviewer is correct that the embryo shown was stage 14. RFP expression from the sensor is first detected at stage 12 and we have replaced the image in Figure 2 to reflect that. Onset of GFP expression in the salivary gland is later and so a stage 17 embryo is also shown in Figure 2.

*At this stage, significant ongoing cell death, as detected with Acridine orange, TUNEL or cCP3, has been reported. Why is there so little RFP in the embryo?*

The reviewer is correct that there is a lot of ongoing cell death at this stage. Very few RFP-positive cells were evident in the single plane confocal image that was provided to show the salivary gland optimally. In Figure 2” we now show a maximum intensity projection of a z-stack of images, which captures cells in more focal planes and it is clear that more RFP-positive cells are evident.

*The Dronc null and p35 experiments are excellent controls showing that the reporter is caspase dependent, but does it somehow miss earlier caspase activation?*

We agree that the Dronc and p35 controls are essential. We don't think there is any reason why the sensor should miss earlier activation. It is clearly expressed in embryos, as shown in the supplement. As mentioned above, very few RFP-positive cells were evident in the single plane confocal image shown in the original image. In Figure 2, we now show a maximum intensity projection of a z-stack of images, which captures cells in more focal planes, and it is clear that quite a few RFP-positive cells are evident. These cells appear to die eventually because we detect virtually no GFP in the embryo outside of the salivary gland. So cells that activate caspase in the embryo virtually all die. This is an interesting observation that suggests that sensitivity to caspase changes during development.

*The wing size experiment shown in Figure 8 could suggest that caspase activity contributes in a positive way to normal wing size. It is important to show that other transgenes expressed from this driver do not suppress wing growth. The driver alone is not a sufficient control. Also the graph exaggerates the wing size difference by starting at 4.2 rather than 0. The wing size should be normalized to another tissue, to account for differences in body size.*

To address the reviewer’s concern, we crossed enGal4 to UAS-p35 and to *w1118*. Since engrailed is only expressed in the posterior compartment of the wing, this allowed us to compare the size of the posterior compartment expressing p35 to the size of the anterior compartment not expressing p35. We present this result in a revised version of Figure 8. The small but reproducible and significant difference was evident in this experiment as well.

[Editors’ note: the author responses to the re-review follow.]

*The manuscript is much improved but there are some remaining issues that need to be addressed before acceptance, as outlined below: The reviewers strongly feel, and you will likely agree, that it is critical to make sure that the caspase being activated in your system, and detected by the sensor is an apoptotic caspase rather than one related to immune response. Although the p35 experiment is valuable, the reviewers have pointed out that it will be important to show whole larval expression, particularly the gut which is quite active in its immune capability. Also, it is rather easy to demonstrate, by placing the sensor directly in a dredd mutant background, to rule out the possibility that this pathway might be involved. Thus, please include the image of the whole larva in p35 and also show sensor expression in a dredd mutant background (X-chromosome location of dredd makes this an easy cross to score in larvae).*

We have crossed the sensor into the dredd mutant, as requested. We do not detect significant differences in sensor activation between wild type and dredd mutants in any larval or adult tissue we examined. These data are now shown in Figure 4—figure supplement 3.

*“Although the p35 experiment is valuable, the reviewers have pointed out that*

*it will be important to show whole larval expression, particularly the gut, which is quite active in its immune capability.”*

In the experiment presented in the paper, p35 was expressed only posterior to the morphogenetic furrow in the eye imaginal disc via the GMR enhancer. Therefore, showing the whole larva would not be informative. It is not possible to express p35 globally together with the CasExpress sensor using existing fly lines or reagents. CasExpress uses Gal4 to drive expression of downstream markers such as RFP in a caspase-dependent manner. Therefore, it is not possible to use Gal4/UAS to drive expression of p35 throughout the larva. This is why we used GMR-p35 to drive expression specifically behind the morphogenetic furrow of the eye imaginal disc. Showing the whole larva would not be informative due to the restricted expression of p35. We are unaware of any existing transgene expressing p35 directly from a ubiquitous promoter, likely because this would be dominant lethal anyway.